# Antimicrobial Gold Nanoclusters: Recent Developments and Future Perspectives

**DOI:** 10.3390/ijms20122924

**Published:** 2019-06-14

**Authors:** Sibidou Yougbare, Ting-Kuang Chang, Shih-Hua Tan, Jui-Chi Kuo, Po-Hsuan Hsu, Chen-Yen Su, Tsung-Rong Kuo

**Affiliations:** 1International Ph.D. Program in Biomedical Engineering, College of Biomedical Engineering, Taipei Medical University, Taipei 11031, Taiwan; d845107003@tmu.edu.tw; 2School of Biomedical Engineering, College of Biomedical Engineering, Taipei Medical University, Taipei 11031, Taiwan; b812105032@tmu.edu.tw (T.-K.C.); b812105028@tmu.edu.tw (S.-H.T.); b812105002@tmu.edu.tw (J.-C.K.); b812106028@tmu.edu.tw (P.-H.H.); willysu683@gmail.com (C.-Y.S.); 3Graduate Institute of Nanomedicine and Medical Engineering, College of Biomedical Engineering, Taipei Medical University, Taipei 11031, Taiwan

**Keywords:** gold nanoclusters, antimicrobial agent, small molecule, macromolecule, antimicrobial mechanism

## Abstract

Bacterial infections have caused serious threats to public health due to the antimicrobial resistance in bacteria. Recently, gold nanoclusters (AuNCs) have been extensively investigated for biomedical applications because of their superior structural and optical properties. Great efforts have demonstrated that AuNCs conjugated with various surface ligands are promising antimicrobial agents owing to their high biocompatibility, polyvalent effect, easy modification and photothermal stability. In this review, we have highlighted the recent achievements for the utilizations of AuNCs as the antimicrobial agents. We have classified the antimicrobial AuNCs by their surface ligands including small molecules (<900 Daltons) and macromolecules (>900 Daltons). Moreover, the antimicrobial activities and mechanisms of AuNCs have been introduced into two main categories of small molecules and macromolecules, respectively. In accordance with the advancements of antimicrobial AuNCs, we further provided conclusions of current challenges and recommendations of future perspectives of antimicrobial AuNCs for fundamental researches and clinical applications.

## 1. Introduction

Treatment of bacterial infection is facing challenge against antimicrobial resistance [1,2,3,4]. The antimicrobial resistance in bacteria remains growing for many reasons included overuse and misuse of antibiotics and the spread of bacteria by various routes [5,6,7]. Therefore, the issue of antimicrobial resistance constitutes a serious risk to public health. According to previous study, the threat of antimicrobial resistance will represent the first cause of death with around ten million per year in 2050 [8,9]. Among different solutions to overcome the antimicrobial resistance, the developments of new antimicrobial agents are critically needed [10,11,12]. Nanomaterials with large surface area and facile functionalization have exhibited superior physical and chemical properties for applications in catalysis, electronic and medicine [13,14,15,16,17,18,19,20]. Recently, organic and inorganic nanomaterials offer an alternative approach to treat infectious diseases caused by bacteria [21,22,23,24,25,26,27]. The antibacterial mechanisms of nanomaterials have been demonstrated, such as the binding between nanomaterials and bacteria for bacterial membrane disruption, photothermal heat generation to kill bacteria by light irradiation onto nanomaterials, photocatalytic production of reactive oxygen species (ROS) via nanomaterials and release of metals ions from nanomaterials to disrupt cellular components of bacteria [28,29,30,31,32,33].

Recent advancements have been focused on the utilizations of metal nanoclusters including gold, silver and copper as the antibacterial agents for bacterial infections [34,35,36,37,38]. Among the metal nanoclusters, gold nanoclusters (AuNCs) have exhibited unique optical and structural properties for the biomedical applications in imaging, detection, and therapy [39,40,41,42,43,44,45]. For the application in therapy, AuNCs conjugated with various surface ligands have been extensively applied as the antimicrobial agents owing to their high biocompatibility, polyvalent effect, easy modification and photothermal stability [46,47,48,49,50,51,52,53,54]. The ligands of amino acids, peptides, antibiotics, antibodies, enzymes, DNA and so forth have been demonstrated for the syntheses of AuNCs [55,56,57,58,59,60,61,62,63,64,65,66]. In this review, we focus on recent achievements dealing with antimicrobial activity of AuNCs capped by various ligands. The ligands embracing different chemical structures were grouped into small molecules (<900 Daltons) and macromolecules (>900 Daltons) [67,68]. In molecular biology and pharmacology, small molecules are commonly defined for the organic compound with the molecular weight lower than 900 Daltons. Small molecules can be used to regulate a biological process [69]. Due to the small size, small molecules are able to penetrate across cell membranes to reach targets in the bacterial cell. In contrast to small molecules, macromolecules (>900 Daltons) are complex and usually exhibit therapeutic effect [70]. Therefore, in this review, AuNCs are classified by their surface ligands included small molecules and macromolecules for the explanation of the antibacterial mechanism of AuNCs (Table 1). The details of ligand size effects and antibacterial mechanisms of ligand-protected AuNCs are also discussed in this review. Finally, challenges and perspectives about antimicrobial AuNCs are provided.

## 2. Small Molecule-Conjugated AuNCs

In recent years, small molecules containing thiol, amine and hydroxyl groups have been used as the ligands to synthesize AuNCs because of low cost, easy accessibility and facile modification. These AuNCs have revealed promising potential as the antimicrobial agents. For example, AuNCs protected by 6-mercaptohexanoic acid (MHA-AuNCs) have been prepared and used as an antimicrobial agent [71]. Zheng et al. have compared the antimicrobial activities of MHA-conjugated gold nanoparticles (MHA-AuNPs), MHA-AuNCs and Au(I)-MHA complexes for Gram-positive *Staphylococcus aureus (S. aureus)*. After incubation with *S. aureus*, MHA-AuNCs have shown superior bacterial killing efficiency ∼95% of the S. aureus. In comparison with MHA-AuNCs, the bacterial killing efficiencies of MHA-AuNPs and Au(I)-MHA complexes are ∼3% and ∼5% for S. aureus, respectively. For the Gram-negative type Escherichia coli (*E. coli*), the bacterial killing efficiencies of MHA-AuNCs, MHA-AuNPs and Au(I)-MHA complexes are individually ∼96%, ∼2% and ∼3%. Herein, MHA-AuNPs have shown no significant antimicrobial activity. However, gold nanoparticles can be used as an antibiotic carrier. The gold nanoparticles with large surface area allow them to conjugate a large number of antibiotics for efficiently against various strains of bacteria [84,85,86,87]. In comparison with gold nanoparticles, the antimicrobial activity of MHA-AuNCs is attributed to their ultra-small size for the improvement of interaction with bacteria. After the internalization of MHA-AuNCs in bacteria, the interaction between MHA-AuNCs and bacteria could cause a metabolic imbalance to result in the increase of intracellular ROS production to eventually kill bacteria (Figure 1) [88].

DNA nanopyramid (DP) is one of DNA nanostructures used in nanomedicine as delivery carrier [89]. Setyawati et al. have used DP as the scaffold to incorporate glutathione-protected AuNCs and Actinomycin D (AMD) to form a nanotheranostic agent (DPAu/AMD) as shown in Figure 2 [72]. The nanotheranostic agent of DPAu/AMD has been applied against *E. coli* and S. aureus. The result indicates that DPAu/AMD show a significant killing efficiency compared to that of the free AMD treatment for both of *E. coli* and *S. aureus*. The DPAu/AMD improve antibacterial effect by reduction of 65% of *S. aureus* population compared to that of 42% for the free AMD. For *E. coli*, the bacterial reductions of DPAu/AMD and free AMD are 48% and 14%, respectively. In comparison with free AMD, the high antibacterial effect of DPAu/AMD can be attributed to that the optimal radius of DPAu/AMD (38.3 nm) can increase the cell uptake for bacteria [90,91].

Sinha et al. have developed a one-pot fabrication with properties including simple, novel, green, economic, environment friendly and convenient for preparation of AuNCs with *Allium cepa L*. (AcL) conjugation [73]. The peel extraction of AcL has biomolecules such as flavonoids, carbohydrates, saponins, amino acid cysteine, sulphoxides, γ-glutamyl peptides and vitamins. The biomolecules with thiol groups in the peel extraction of AcL have been used to reduce the precursor of Au (III) to form Au (I) and Au (0) for the formation of AuNCs [92]. In this work, the antibacterial activities of AuNCs, AcL and Tetracycline antibiotic have been investigated against Gram-negative *E. coli*. Results show that AuNCs have the highest bacterial killing efficiency, followed by Tetracycline antibiotic and then extraction of AcL the least. The highest antibacterial activity of AuNCs can be attributed to that the large surface area and easy penetration ability of AuNCs can increase the interaction between AuNCs and bacterial membrane to result in the death of bacteria. To combat bacteria, water-soluble biofunctional AuNCs conjugated with mannose (Man-AuNCs) have been developed for the sensitive and selective detection and bacterial inhibition of *E. coli*. (Figure 3) [74]. The mannose ligands conjugated on the surfaces of Man-AuNCs have induced strong multivalent interactions between Man-AuNCs and FimH proteins located on the bacterial pili of *E. coli*. The bacterial aggregations caused by Man-AuNCs lead to the inhibition of the growth of *E. coli*. The antibacterial activity of Man-AuNCs has been shown in Figure 3. The growth curve of *E. coli* in sterile LB media has shown a very low growth rate of *E. coli* after incubated with Man-AuNCs (>250 nM) as shown in Figure 3A. In Figure 3B, the number of colonies on the LB agar plates of untreated and Man-AuNCs-treated *E. coli* have been calculated to be 78 and 18 colony-forming unit (CFU), respectively. In this work, the Man-AuNCs have great potential for use as an antibacterial agent due to high ligand density of mannose on the surface of Man-AuNCs for multivalent interactions with *E. coli*.

Recently, Xie et al. have synthesized and functionalized AuNCs using positive ligands including quaternary ammonium (QA-AuNCs), nona-arginine peptide (R9-AuNCs) and the transactivator of transcription peptide (Tat-AuNCs) by one-pot synthesis with glutathione as the reductant [75]. The antibacterial activities of AuNCs have been investigated by measuring their minimal inhibitory concentrations (MICs) in Gram-positive *S. aureus*, methicillin-resistant *Staphylococcus aureus* (MRSA), Gram-negative *E. coli* and multidrug-resistant *E. coli*. With the changes of ligand/reductant (L/R) ratio, the QA-AuNCs with an L/R ratio of 0.5:1 has exhibited superior antibacterial effect for the four targeting bacteria (Figure 4). The antibacterial mechanism of QA-AuNCs can be ascribed to the fact that the positive charge on the surface of QA-AuNCs can promote electrostatic adsorption onto bacterial cell membrane with negative charge. Additionally, then QA-AuNCs have induced disruption of membrane integrity, increase of membrane permeability and dissipation of the membrane potential of *S. aureus*. Eventually, QA-AuNCs can improve the generation of ROS and cause the death of bacteria [93]. Overall, QA-AuNCs have shown promising potential as the antibacterial agent using physicochemical mechanism for the skin infection model and the bacteremia model caused by MRSA [9,94].

Moreover, four ligands which are analogues of mercaptopyrimidine including 4-amino-2-mercaptopyrimidine (AMP), 4,6-diamino-2-mercaptopyrimidine (DAMP), 4-amino-6-hydroxyl-2-mercaptopyrimidine (AHMP), and 4,6-dihydroxyl-2-mercaptopyrimidine (DHMP) have been used to synthesize mercaptopyrimidine conjugated AuNCs to combat multidrug-resistant bacteria [76]. For these AuNCs, DHMP-conjugated AuNCs (AuDHMP) have exhibited negative charge and the others AuNCs of AuAMP, AuDAMP and AuAHMP have shown positive charges. The zeta potentials for AuDHMP, AuAMP, AuDAMP and AuAHMP are −38.6 ± 1.8, +33.6 ± 1.4, +37.6 ± 1.1 and +12.7 ± 0.7 mV, respectively. All AuNCs have revealed antimicrobial activities against *E. coli* ATCC 35218 (Gram-negative bacteria) and S. aureus ATCC 29213 (Gram-positive bacteria). The AuDAMP have the best performance of antimicrobial activity compared to AuAMP, AuAHMP and AuDHMP because the high positive surface charge of AuDAMP can facilitate their electrostatic adsorption onto the surface of bacteria to increase internalization of AuDAMP into bacteria. Furthermore, AuDAMP also can fight mutli-drug resistant bacteria such as *E. coli*, *Acinetobacter baumannii (A. baumannii)*, *Pseudomonas aeruginosa*, *Klebsiella pneumonia (K. pneumonia)*, methicillin-resistant *Staphylococcus aureus (MRSA)* and vancomycin-resistant *Enterococcus faecium (E. faecium).* To kill bacteria, the mechanisms of antimicrobial AuDAMP have been demonstrated by the combination of cell membrane destruction, DNA damage and ROS generation caused by AuDAMP to bacteria (Figure 5) [95,96,97].

Nanomaterial-based antimicrobial agents with positive surface charges are generally considered to lead higher antimicrobial activities as shown in the example of AuDAMP. However, Zheng et al. have prepared five types of Au_25_NCs with negative surface charges including Au_25_NCs protected by MHA (Type I), Au_25_NCs protected by p-mercaptobenzoic acid (MBA) (Type II), Au_25_NCs protected by cysteine (Cys) (Type III), Au_25_NCs protected by MHA and cysteamine (Cystm) (Type IV) and Au_25_NCs protected by MHA and 2-mercaptoethanol (AuMetH) (Type V) [77]. By the designs of surface ligands, Au_25_NCs with more negative surface charges on the surface could induce more ROS generation to react with metabolic enzyme of bacteria and then to kill the bacteria (Figure 6) [98,99]. The results in this work indicate that surface charge of AuNCs plays a pivotal role in antimicrobial properties.

## 3. Macromolecule-Conjugated AuNCs

Macromolecules are also commonly used as the surface ligands to prepare AuNCs for antibacterial applications. With the conjugations of macromolecules, AuNCs have shown various antibacterial effects. Recently, Chen et al. have synthesized lysozyme capped AuNCs (lysozyme-AuNCs) as an antimicrobial agent [78]. The enzyme of lysozyme can hydrolyze the cell walls of pathogenic bacteria [100,101,102]. The lysozyme-AuNCs have exhibited bacteriostatic effects against pan-drug-resistant *Acinetobacter baumannii* (*A. baumannii*) and vancomycin-resistant *Enterococcus faecalis* (*E. faecalis*) because of the multivalent interactions of the Lysozyme-AuNCs with the target bacteria. Furthermore, lysozyme conjugated AuNCs have been functionalized with ampicillin (AuNC-L-Amp) to combat MRSA and other non-resistant bacteria [79]. In this work, AuNC-L-Amp have been proved to overcome the increased β-lactamase at the site of MRSA and then the multivalent binding of AuNC-L-Amp onto the bacterial surface can be applied to enhance the permeation of AuNC-L-Amp into bacteria. The AuNC-L-Amp have shown a significant enhancement (50–89% fold increase) of antimicrobial activity compared to that of free-Amp for nonresistant bacterial pathogens. The AuNC-L-Amp have also revealed antimicrobial activity for MRSA, but free-Amp and AuNC-L have exhibited no significant antimicrobial activity for MRSA (Figure 7). The mechanism for the use of AuNC-L-Amp as the antimicrobial agent can be ascribed to the reasons including the increase of Amp concentration in bacteria, multivalent presentation of antibiotics, hydrolysis of cell wall by lysozyme, dysfunction of the bacterial efflux pump and ions released from AuNCs to inhibit bacterial growth [103,104,105,106].

Antibiotic of vancomycin for all Gram-positive bacteria has been used as a template and reducing agent to synthesize vancomycin-bound AuNCs (AuNC@Van) [80]. Antibacterial ability of AuNC@Van has been evaluated on Gram-negative *E. coli* and Gram-positive *S. aureus.* The results indicate that AuNC@Van has good antimicrobial activity for both *E. coli* and *S. aureus.* To further study the antibacterial mechanism of AuNC@Van, Liang et al. have investigated the morphological changes of *E. coli* and *S. aureus* incubated with AuNC@Van by SEM. After incubation with AuNC@Van for 48 h, the bacterial cell wall has shown wrinkle and destruction. Afterward, because of the damage of bacterial cell wall, vancomycin on the surface of AuNC@Van can easily penetrate into bacteria to enhance its antimicrobial activity. Moreover, Li et al. have chosen a pentapeptide γ-ECG_D_A_D_A (GSHaa) to prepare AuNCs (Au-SGaa, SGaa denotes dehydrogenated GSHaa) and then Au-SGaa have been conjugated with vancomycin (Au-SGaa-Van) [81]. In this study, the Gram-positive bacteria *S. aureus* and Gram-negative bacteria *E. coli* have been selected to assess antibacterial activity of Au-SGaa-Van. As shown in Figure 8, Au-SGaa-Van and vancomycin have shown significant antibacterial effect against *S. aureus*. However, there is no antibacterial activity of Au-SGaa-Van for *E. coli*. The results indicate that antibacterial activity of the Au-SGaa-Van is caused by the vancomycin on their surface.

Furthermore, Liao et al. have constructed AuNCs to inhibit endotoxin activity by blocking on active site of lipopolysaccharide (LPS) [82]. LPS is one of constituents of Gram-negative bacteria responsible of sepsis to humans [107]. They have decorated subnanometer gold clusters (SAuNCs) using methyl and ethyl groups to synthesize SAuNC-M and SAuNC-E, respectively. Additionally, hydrophilic SAuNCs (SAuNC-A) and hydrophobic SAuNCs (SAuNC-H) have been synthesized. The SAuNC-M and SAuNC-E have caused the inhibition of LPS aggregation but SAuNC-A and SAuNC-H have been validated to produce LPS aggregation [108]. The endotoxin activity can be effectively blocked by SAuNCs including SAuNC-M and SAuNC-E as means to fight sepsis. Results of their work have shown that the antiendotoxin SAuNC-M and SAuNC-E could be the efficacious antimicrobial agents to prevent sepsis due to infection from Gram-negative bacteria (Figure 9).

The bacitracin-directed silver, gold and copper nanoclusters (AgNCs@Bacitracin, AuNCs@Bacitracin and CuNCs@Bacitracin) have been obtained by Wang and coworkers [83]. The antibacterial activities of these nanoclusters have been investigated by the use of *S. aureus*. The antibacterial mechanism of these nanoclusters have been demonstrated with the coordination between bacitracin and the metallic atoms. In this work, the AgNCs@Bacitracin, AuNCs@ Bacitracin and CuNCs@Bacitracin have respectively revealed 72.3%, 26.6% and 30.5% of the damage of bacterial cell wall. Furthermore, AgNCs@Bacitracin, AuNCs@Bacitracin and CuNCs@Bacitracin have caused the increases of the intracellular ROS production leading to the bacterial death (Figure 10). Taking the advantages together, the nanoclusters of AgNCs@Bacitracin, AuNCs@Bacitracin and CuNCs@Bacitracin have shown superior antibacterial activities because of the damage of bacterial cell wall and the increase of intracellular ROS production. Additionally, bacitracin on the surface of nanoclusters has also cooperated with metallic atoms to improve antibacterial activity of nanoclusters. Among these three nanoclusters, AgNCs@Bacitracin have shown the best antibacterial activity compared to that of AuNCs@Bacitracin and CuNCs@Bacitracin. Although sliver-based nanoclusters have exhibited higher antibacterial activity compared to that of gold-based nanoclusters, gold-based nanoclusters are still the most promising metallic antibacterial agent due to their remarkable advantages such as high biocompatibility, polyvalent effect, easy modification and photothermal stability.

## 4. Challenges and Opportunities

In this mini review, we have summarized recent achievements of AuNCs conjugated with small molecules and macromolecules for the applications as the antimicrobial agents (Table 1). These studies have demonstrated that AuNCs can be the potential antimicrobial agents because of their high biocompatibility, polyvalent effect, easy modification and photothermal stability. Although different AuNCs have been proven as the antimicrobial agents, however, their antimicrobial activities still need to be improved. The first challenge to improve the antimicrobial activity of AuNCs is to prepare AuNCs conjugated with antimicrobial surface ligands. The antimicrobial activity can be enhanced by the use of synergistic effect between AuNCs and antimicrobial surface ligands. The second challenge for antimicrobial AuNCs is to increase their cell uptake. With the controls of surface ligands, AuNCs can bear positive charge and negative charge and even to have target-specific property for bacteria to increase the cell uptake of AuNCs. The third challenge for antimicrobial AuNCs is to investigate the details of antimicrobial mechanisms in bacteria. Until now, there are various mechanisms to explain the antimicrobial performance of AuNCs. Therefore, experimental and theoretical investigations of the metabolisms of AuNCs in bacteria are still required for better understanding their antimicrobial activity. Overall, to realize the antimicrobial agents of AuNCs, a lot of work still need to be completed for the improvement of antimicrobial activity of AuNCs to meet the requirement in clinic application. With extensive investigations, we believe that AuNCs can be applied as the significant antimicrobial agents in clinic in the near future.

## Figures and Tables

**Figure 1 ijms-20-02924-f001:**
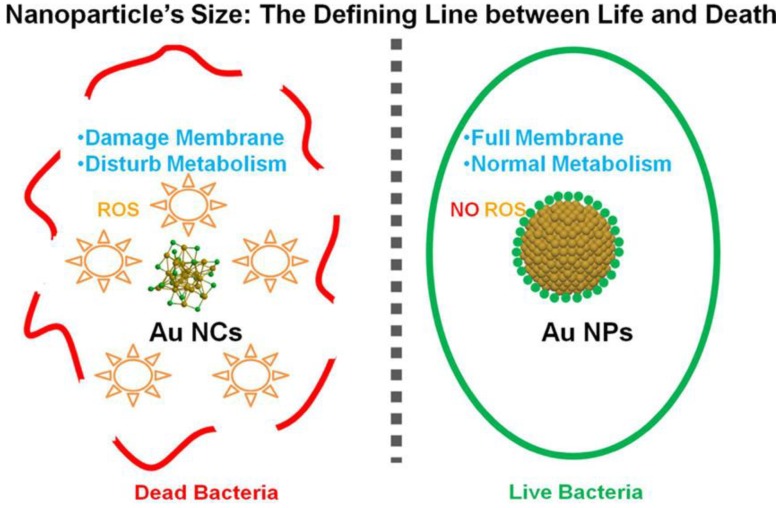
Antimicrobial activity of MHA-AuNCs due to the increase of interaction between MHA-AuNCs and bacteria. Reproduced with permission from Reference [71]. Copyright © 2017, American Chemical Society.

**Figure 2 ijms-20-02924-f002:**
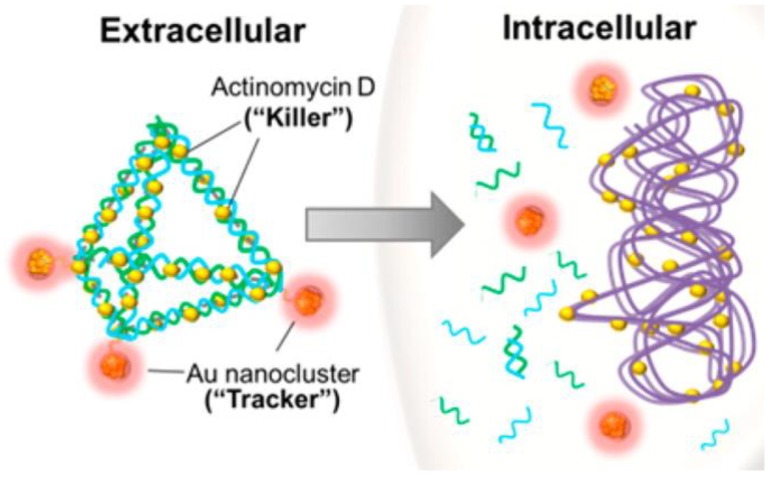
Representative scheme of DPAu/AMD as a nanotheranostic agent. Reproduced with permission from Reference [72]. Copyright © 2014, American Chemical Society.

**Figure 3 ijms-20-02924-f003:**
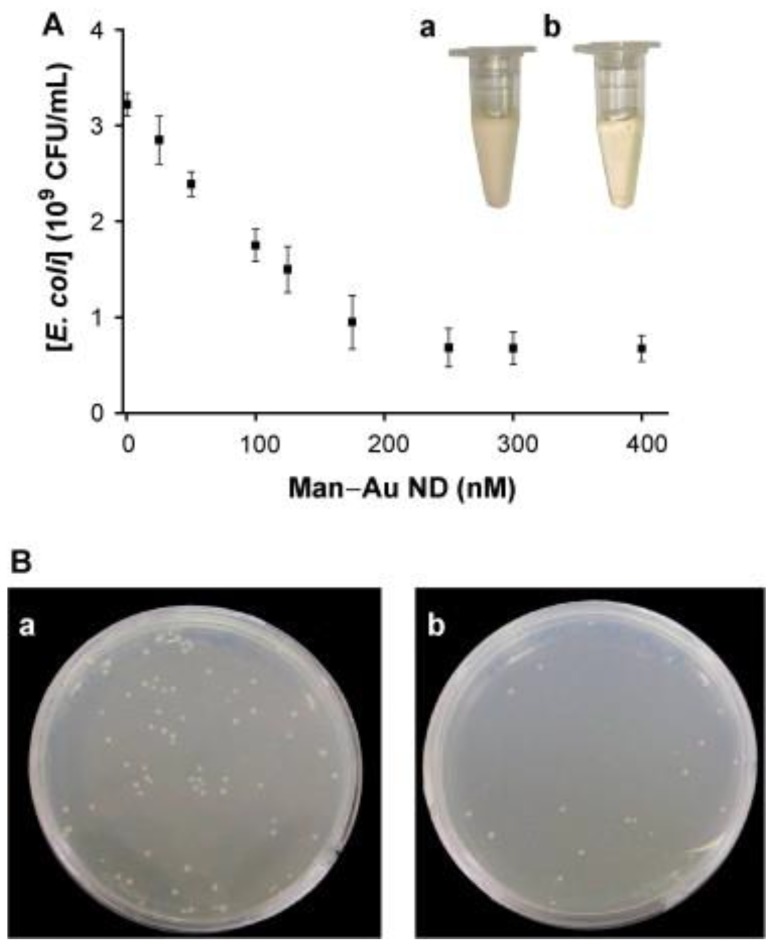
(**A**) Antibacterial effect of Man-AuNCs with their concentrations from 0 to 400 nM. (**B**) Colony formation of *E. coli* on LB agar plates in the (a) absence and (b) presence of Man-AuNCs (250 nM). Insets of Figure 3A indicate photographs of *E. coli* (1.0×10^8^ CFU/mL) grown for 10 h in the LB medium in the (a) absence and (b) presence of Man-AuNCs (250 nM). Reproduced with permission from Reference [74]. Copyright © 2011, Elsevier.

**Figure 4 ijms-20-02924-f004:**
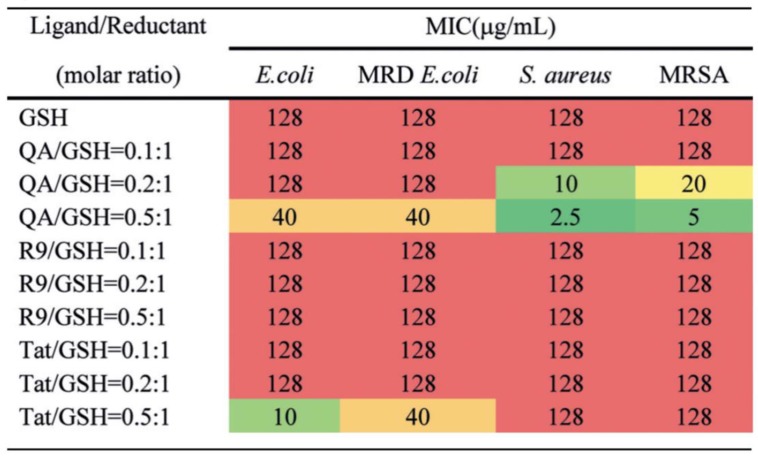
Antibacterial activities of AuNCs conjugated with different ligands by measuring their MICs. The lower MIC of AuNCs show higher antibacterial activity. Reproduced with permission from Reference [75]. Copyright © 2018, WILEY-VCH Verlag GmbH & Co. KGaA, Weinheim.

**Figure 5 ijms-20-02924-f005:**
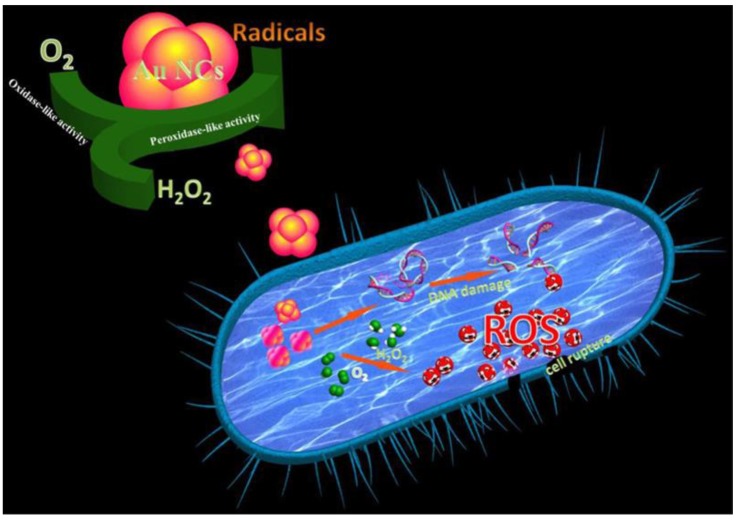
The mechanisms of cell membrane destruction, DNA damage and ROS generation for AuDAMP to kill bacteria. Reproduced with permission from Reference [76]. Copyright © 2018, American Chemical Society.

**Figure 6 ijms-20-02924-f006:**
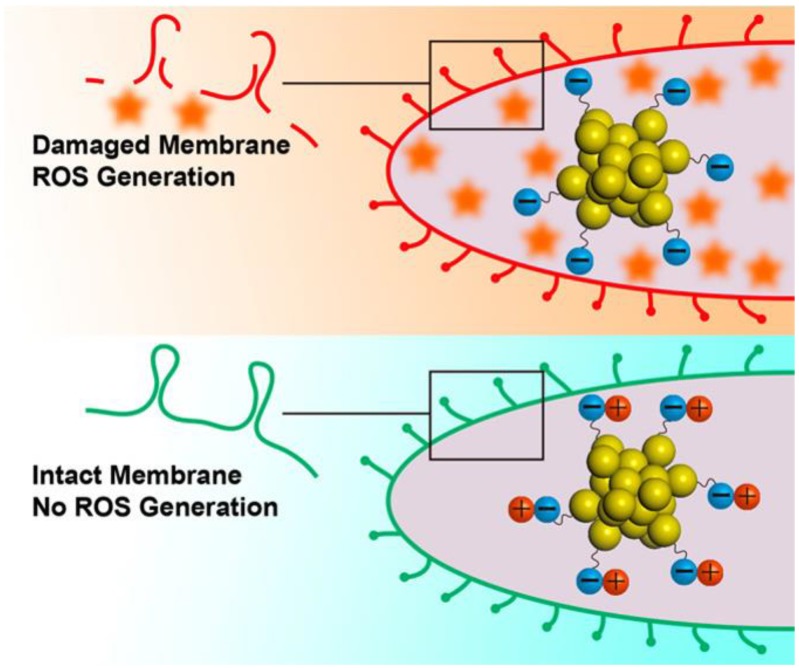
Surface ligand chemistry of AuNCs could determine their antimicrobial ability. Reproduced with permission from Reference [77]. Copyright © 2018, American Chemical Society.

**Figure 7 ijms-20-02924-f007:**
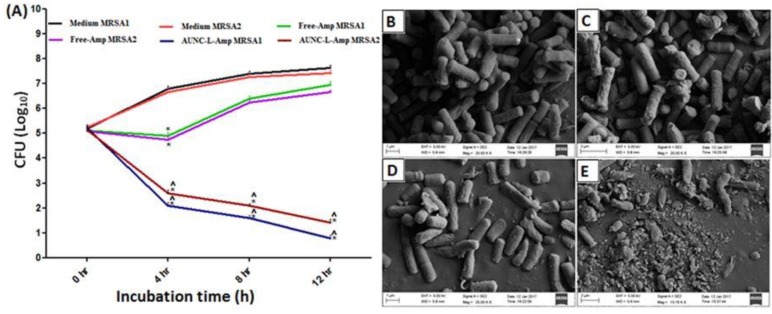
(**A**) Antimicrobial activities of AuNC-L-Amp and free-Amp at different incubation times for MRSA1 and 2. SEM images of MRSA incubation with (**B**) PBS, (**C**) AuNC-L, (**D**) Free-Amp and (**E**) AuNC-L-Amp. SEM images indicate that AuNC-L and free-Amp did not cause the changes of bacterial morphology. On the other hand, AuNC-L-Amp induced the cellular structure of MRSA. Reproduced with permission from Reference [79]. Copyright © 2018, Springer Nature.

**Figure 8 ijms-20-02924-f008:**
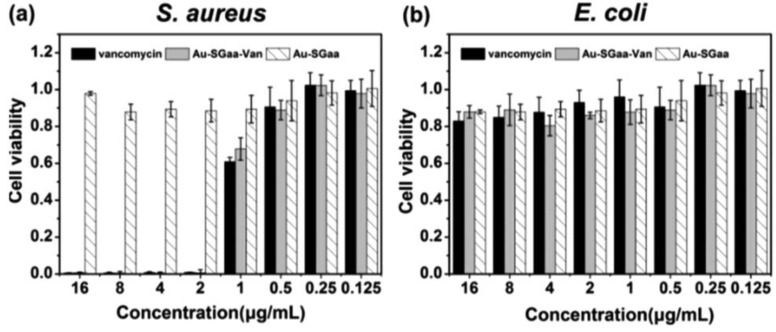
(**a**) Antibacterial activity of Au-SGaa-Van, vancomycin and Au-SGaa against Gram-positive *S. aureus*. (**b**) Antibacterial activity of Au-SGaa-Van, vancomycin and Au-SGaa against Gram-negative *E. coli*. Reproduced with permission from Reference [81]. Copyright © 2018, Royal Society of Chemistry.

**Figure 9 ijms-20-02924-f009:**
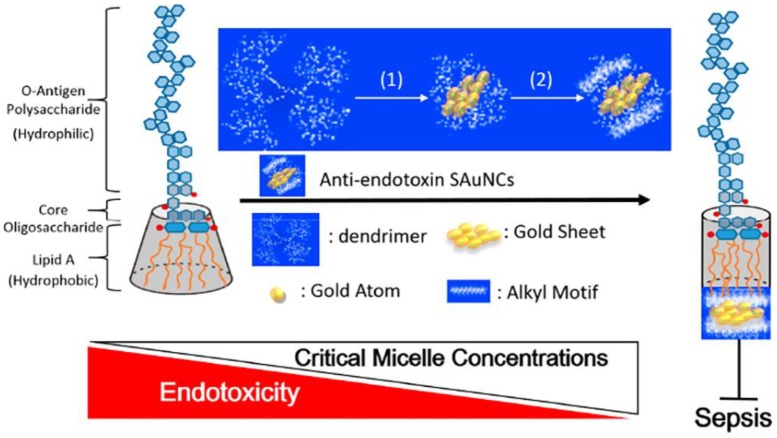
Illustration of interaction between SAuNCs, lipid A of LPS and sepsis progression. Reproduced with permission from Reference [82]. Copyright © 2018, American Chemical Society.

**Figure 10 ijms-20-02924-f010:**
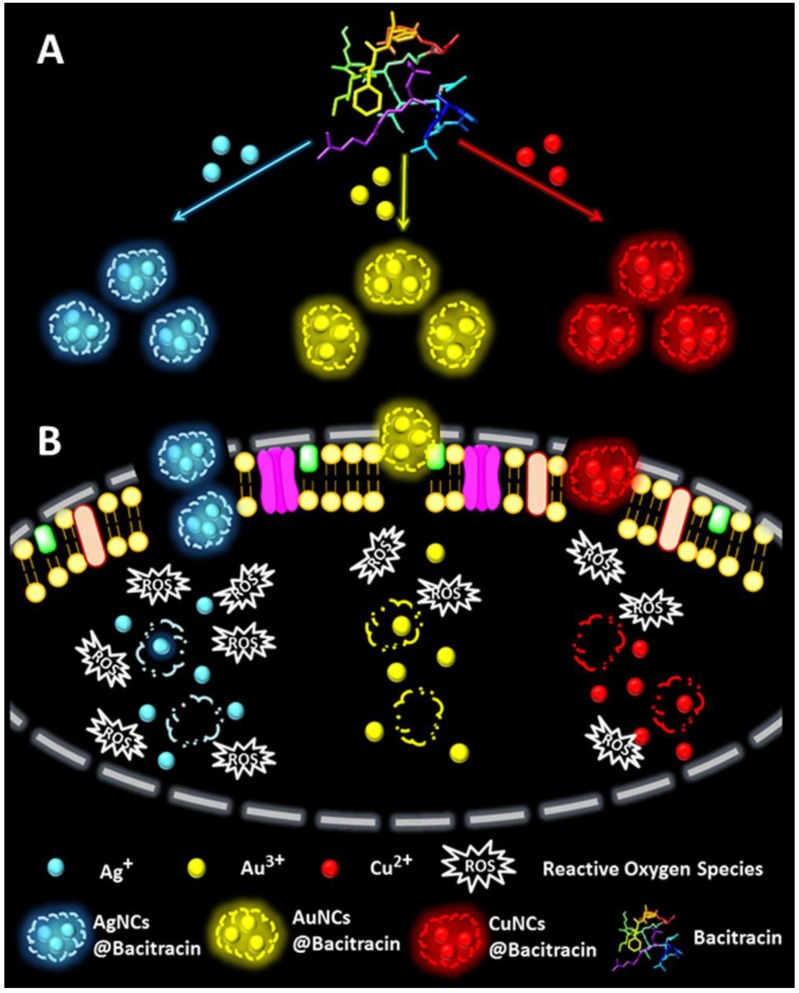
(**A**) Illustration of preparations of AgNCs@Bacitracin, AuNCs@Bacitracin, and CuNCs@Bacitracin. (**B**) Bacteria incubated with AgNCs@Bacitracin, AuNCs@Bacitracin, or CuNCs@Bacitracin. Reproduced with permission from Reference [83]. Copyright © 2019, American Chemical Society.

**Table 1 ijms-20-02924-t001:** Ligands and antibacterial mechanisms of AuNCs in this review.

Types	Ligands (Molecular Weight)	Antibacterial Mechanisms	References
Small molecules	6-Mercaptohexanoic acid (148 Da)	Increase of ROS generation by MHA-AuNCs to kill bacteria	[71]
	Glutathione (307 Da)	Optimal radius of DPAu/AMD for the increase of cell uptake	[72]
	*Allium cepa L*. (Mixture)	Increase of the interaction between AuNCs and bacterial membrane	[73]
	Mannose (180 Da)	Bacterial aggregations	[74]
	Quaternary ammonium (282 Da)	Increase of ROS generation by QA-AuNCs with positive charge	[75]
	4-Amino-2-mercaptopyrimidine (127 Da)	Increase of ROS generation by AuDHMP with positive charge	[76]
	6-Mercaptohexanoic acid (148 Da)	Increase of ROS generation by Au_25_NCs with negative charge	[77]
Macromolecules	Lysozyme (143000 Da)	Hydrolysis of bacterial cell wall by lysozyme-AuNCs	[78]
	Lysozyme (143000 Da)	Multivalent interactions between AuNC-L-Amp and bacterial	[79]
	Vancomycin (1449 Da)	Delivery of vancomycin into bacteria by AuNC@Van	[80]
	Vancomycin (1449 Da)	Delivery of vancomycin into bacteria by Au-SGaa-Van	[81]
	G_4_NH_2_ & G_4_OH (14266 & 14277 Da)	Inhibition of LPS aggregation	[82]
	Bacitracin (1422 Da)	Damage of cell wall and increase of ROS production by AuNCs@Bacitracin	[83]

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
