# Peer review of "Antimicrobial Gold Nanoclusters: Recent Developments and Future Perspectives"

_ijms, 2019, doi:10.3390/ijms20122924_

Reviewer 1 Report

The goal of this mini-review is to provide an authoritative report of the recent achievements for the utilization of AuNCs as the antimicrobial agents, and thus this review appears useful for the protected metal nanocluster community.  Note also that 22/102 references are from 2017-2018 and even 3 references are from 2019.

Overall there is nothing wrong or inaccurate about this review, but it is not very appealing. The mini-review is a kind of enumeration of different examples (displayed in Table I), where the guiding principle is missing.

In this review, the authors have classified the antimicrobial AuNCs by their surface ligands included small molecules (< 900 Daltons) and macromolecules (> 900 Daltons). It is an interesting approach but this frontier at ~900 Da needs to be justified in terms of the different physico-chemical properties in determining their antimicrobial efficacy (e.g. in terms of damages to bacterial membrane, and/or to subcellular structures).

For instance a short section entitled “Ligands size effects and antibacterial mechanisms of ligand-protected AuNCs” should be added after the introduction allowing to pave this classification (< and > 900 Da). Schematics summarizing the different antibacterials mechanisms of AuNCs according to this classification (small and large ligands) should be added. Also, Table I presented in conclusion should be put in the short section after introduction. In Table I, please add MW of ligands to better justify this frontier at ~900 Da.

It could be also interesting to have a critical discussion (with additional references) of the superior antibacterial effect of “molecular-like” Au NCs as compared to large “plasmonic” Au nanoparticles, for which the ligand’s roles are different. Also additional references should be incorporated in this discussion on the possible antibacterial superior effect of “molecular-like” Au NCs as compared to other metal nanoclusters (e.g. Ag and Cu) but also mixed Ag/Au nanoclusters.

The impact of the group of Tsung-Rong Kuo in the field AuNCs as the antimicrobial agents should be better highlighted in Introduction.

Author Response

Comment:

The goal of this mini-review is to provide an authoritative report of the recent achievements for the utilization of AuNCs as the antimicrobial agents, and thus this review appears useful for the protected metal nanocluster community. Note also that 22/102 references are from 2017-2018 and even 3 references are from 2019.

Reply to the reviewer:

Gold nanoclusters (AuNCs) are emerging as novel antimicrobial agents for biomedical application in therapy. Moreover, antimicrobial AuNCs have shown high biocompatibility, polyvalent effect, easy modification and photothermal stability. The applications of AuNCs as the antimicrobial agents are just at the beginning stage and many scientific issues have not been discussed yet. We believe that this mini review has shown some new and interesting aspects in the field of antibacterial AuNCs for fundamental investigations and further clinical applications.

Comment:

Overall there is nothing wrong or inaccurate about this review, but it is not very appealing. The mini-review is a kind of enumeration of different examples (displayed in Table I), where the guiding principle is missing. In this review, the authors have classified the antimicrobial AuNCs by their surface ligands included small molecules (< 900 Daltons) and macromolecules (> 900 Daltons). It is an interesting approach but this frontier at ~900 Da needs to be justified in terms of the different physico-chemical properties in determining their antimicrobial efficacy (e.g. in terms of damages to bacterial membrane, and/or to subcellular structures).

Reply to the reviewer:

In molecular biology and pharmacology, small molecules are commonly defined for the organic compound with the molecular weight low than 900 Daltons. Small molecule can be used to regulate a biological process [1]. Due to the small size, small molecules are able to penetrate across cell membranes to reach targets in the bacterial cell. In contrast to small molecules, macromolecules (> 900 Daltons) are complex and usually exhibited therapeutic effect [2]. Therefore, in this review, AuNCs were classified by their surface ligands included small molecules and macromolecules for the explanation of the antibacterial mechanism of AuNCs. The details of ligand size effects and antibacterial mechanisms of ligand-protected AuNCs were also discussed in this review.

References

1.  Nwibo, D. D.; Levi, C. A.; Nwibo, M. I., J. appl. dent. 2015, 14, 70-77.

2.  Hay, M.; Thomas, D. W.; Craighead, J. L.; Economides, C.; Rosenthal, J., Nat. Biotechnol. 2014, 32, 40.

Revision made:

We have added the description to explain the reason why we classified AuNCs by their surface ligands in revised manuscript. We wrote “In molecular biology and pharmacology, small molecules are commonly defined for the organic compound with the molecular weight low than 900 Daltons. Small molecule can be used to regulate a biological process [71]. Due to the small size, small molecules are able to penetrate across cell membranes to reach targets in the bacterial cell. In contrast to small molecules, macromolecules (> 900 Daltons) are complex and usually exhibited therapeutic effect [72]. Therefore, in this review, AuNCs were classified by their surface ligands included small molecules and macromolecules for the explanation of the antibacterial mechanism of AuNCs. The details of ligand size effects and antibacterial mechanisms of ligand-protected AuNCs were also discussed in this review.”

Comment:

For instance a short section entitled “Ligands size effects and antibacterial mechanisms of ligand-protected AuNCs” should be added after the introduction allowing to pave this classification (< and > 900 Da).

Reply to the reviewer:

We thank Reviewer 1 for the suggestion to add a short section to pave the classification of AuNCs conjugated with small molecules and macromolecules. We have a short section to explain the reason for the classification of AuNCs more clearly.

Revision made:

A short section to explain the reason for the classification of AuNCs has been added in the section of “Introduction” in the revised manuscript. We wrote “In molecular biology and pharmacology, small molecules are commonly defined for the organic compound with the molecular weight low than 900 Daltons. Small molecule can be used to regulate a biological process [71]. Due to the small size, small molecules are able to penetrate across cell membranes to reach targets in the bacterial cell. In contrast to small molecules, macromolecules (> 900 Daltons) are complex and usually exhibited therapeutic effect [72]. Therefore, in this review, AuNCs were classified by their surface ligands included small molecules and macromolecules for the explanation of the antibacterial mechanism of AuNCs. The details of ligand size effects and antibacterial mechanisms of ligand-protected AuNCs were also discussed in this review.”

Comment:

Schematics summarizing the different antibacterials mechanisms of AuNCs according to this classification (small and large ligands) should be added.

Reply to the reviewer:

We thank the suggestion from Reviewer 1. The examples of AuNCs with different surface ligands reported in this review have been summarized in Table 1. The related antibacterial mechanisms of those AuNCs have been also provided in Table 1.

Comment:

Also, Table I presented in conclusion should be put in the short section after introduction.

Reply to the reviewer:

We have removed Table 1 to the section at the end of introduction.

Comment:

In Table I, please add MW of ligands to better justify this frontier at ~900 Da.

Reply to the reviewer:

We have revised Table 1 to show the molecular weights of ligands.

Revision made:

The molecular weights of ligands have been added in Table 1.

Comment:

It could be also interesting to have a critical discussion (with additional references) of the superior antibacterial effect of “molecular-like” Au NCs as compared to large “plasmonic” Au nanoparticles, for which the ligand’s roles are different.

Reply to the reviewer:

We appreciate the insightful suggestion for Reviewer 1. We have added a discussion to explain the different roles of surface ligands for antibacterial agents of gold nanoclusters and gold nanoparticles.

Revision made:

We have added a critical discussion to describe the difference between gold nanoclusters and gold nanoparticles for the uses as antibacterial agents on Page 6 in the revised manuscript. We wrote “Herein, MHA-AuNPs have shown no significant antimicrobial activity. However, gold nanoparticles can be utilized as an antibiotic carrier. The gold nanoparticles with large surface area allow them to conjugate a large number of antibiotics for efficiently against various strains of bacteria [86-89]. In comparison with gold nanoparticles, the antimicrobial activity of MHA-AuNCs is attributed to their ultra-small size for the improvement of interaction with bacteria. After the internalization of MHA-AuNCs in bacteria, the interaction between MHA-AuNCs and bacteria could cause a metabolic imbalance to result in the increase of intracellular ROS production to eventually kill bacteria (Figure 1) [90].”

Comment:

Also additional references should be incorporated in this discussion on the possible antibacterial superior effect of “molecular-like” Au NCs as compared to other metal nanoclusters (e.g. Ag and Cu) but also mixed Ag/Au nanoclusters.

Reply to the reviewer:

In this review, the bacitracin-directed silver, gold and copper nanoclusters (AgNCs@Bacitracin, AuNCs@Bacitracin and CuNCs@Bacitracin) obtained by Wang and coworkers have been reported. The antibacterial activities of these nanoclusters have been investigated in bacteria. We believe that this work can be a suitable example to depict the antibacterial differences of metal nanoclusters. In original manuscript, we wrote “The bacitracin-directed silver, gold and copper nanoclusters (AgNCs@Bacitracin, AuNCs@Bacitracin and CuNCs@Bacitracin) have been obtained by Wang and coworkers [85]. The antibacterial activities of these nanoclusters have been investigated by the use of S. aureus. The antibacterial mechanism of these nanoclusters have been demonstrated with the coordination between bacitracin and the metallic atoms. In this work, the AgNCs@Bacitracin, AuNCs@ Bacitracin and CuNCs@Bacitracin have respectively revealed 72.3%, 26.6% and 30.5% of the damage of bacterial cell wall. Furthermore, AgNCs@Bacitracin, AuNCs@Bacitracin and CuNCs@Bacitracin have caused the increases of the intracellular ROS production leading to the bacterial death (Figure 10). Taking the advantages together, the nanoclusters of AgNCs@Bacitracin, AuNCs@Bacitracin and CuNCs@Bacitracin have shown superior antibacterial activities because of the damage of bacterial cell wall and the increase of intracellular ROS production. Additionally, bacitracin on the surface of nanoclusters has also cooperated with metallic atoms to improve antibacterial activity of nanoclusters. Among these three nanoclusters, AgNCs@Bacitracin have shown the best antibacterial activity compared to that of AuNCs@Bacitracin and CuNCs@Bacitracin. Although sliver-based nanoclusters have exhibited higher antibacterial activity compared to that of gold-based nanoclusters, gold-based nanoclusters are still the most promising metallic antibacterial agent due to their remarkable advantages such as high biocompatibility, polyvalent effect, easy modification and photothermal stability.”

Comment:

The impact of the group of Tsung-Rong Kuo in the field AuNCs as the antimicrobial agents should be better highlighted in Introduction.

Reply to the reviewer:

We have provided three papers published from our group related with AuNCs for biomedical applications in imaging, detection, and therapy. In the section of “Introduction”, we wrote “Among the metal nanoclusters, gold nanoclusters (AuNCs) have exhibited unique optical and structural properties for the biomedical applications in imaging, detection, and therapy [40-46]. For the application in therapy, AuNCs conjugated with various surface ligands have been extensively applied as the antimicrobial agents owing to their high biocompatibility, polyvalent effect, easy modification and photothermal stability [47-56]. The ligands of amino acids, peptides, antibiotics, antibodies, enzymes, DNA and so forth have been demonstrated for the syntheses of AuNCs [57-68].”

Three references published by our group as following:

45.  Cheng, T. M.; Chu, H. L.; Lee, Y. C.; Wang, D. Y.; Chang, C. C.; Chung, K. L.; Yen, H. C.; Hsiao, C. W.; Pan, X. Y.; Kuo, T. R.; Chen, C. C., Anal. Chem. 2018, 90, 3974-3980.

46.  Kaur, N.; Aditya, R. N.; Singh, A.; Kuo, T. R., Nanoscale Res. Lett. 2018, 13, 302.

57.  Li, C.-H.; Kuo, T.-R.; Su, H.-J.; Lai, W.-Y.; Yang, P.-C.; Chen, J.-S.; Wang, D.-Y.; Wu, Y.-C.; Chen, C.-C., Sci. Rep. 2015, 5, 15675.

Reviewer 2 Report

The authors summarized the recent development of antimicrobial gold nanoclusters and divided them into two categories, namely, small molecules and macromolecules. The citing articles are up to date; however, there are nontrivial number of grammatic errors and typos in this review article. In addition, some information summarized in this review does not clearly convey the ideas from original articles. Please refer to the following suggestions and make proper corrections. The authors should also thoroughly proofread their manuscript before it is ready for publication.

Ref 8 should come from O'Neill J. Review on Antimicrobial Resistance Antimicrobial Resistance: Tackling a crisis for the health and wealth of nations. London: Review on Antimicrobial Resistance. 2014

Line 42,68,84,141,177,200,228,261,307: included should be including.

Line 90-92: “The ligands of amino acids, peptides, antibiotics, antibodies, enzymes, DNA and so forth have been demonstrated for the syntheses of AuNCs.” is unclear. It should be rewritten for clarity.

Line 100: coast should be cost.

Line 130: The definition of “S. aureus burden” is unclear.

Line 147: should be Au (I).

Line 149: have “the” highest

Line 160: “have been show in the Figure 3” should be “have been shown in Figure 3”.

Line 164: define CFU.

In the caption of Figure 3: Insets of Figure 4A should be Figure 3A.

Line 192: provide the references for skin infection model.

Line 204-206: For clarity, it is better to report the charges or Zeta potential for each conjugate as it is important for their antimicrobial activity.

Line 233: The use of “inference” is unclear.

Line 275: temple should be template  

Line 284 and in Table 1: pentapeptide γ-ECGDADA should be γ-ECGDADA, where DADA means D-alanine-D-alanine.

Line 285: SGaa is not defined.

Line 286-289, 320: Bacteria names should be italicized.

In the caption of Figure 10, the word “synthesis” is misleading. It should be changed from “Illustration of synthesis of” to “Illustration of the preparation of”

In the conclusion, the authors introduced some current challenges which are not mentioned in the previous sections. It is generally not a good practice to introduce new ideas in the conclusion. I suggest the authors to write more details about the current challenge in a new section. This will add more valuable information to the readers.

In Table 1, some of the antimicrobial mechanisms are unclear. For example, does “increase of vancomycin” mean the increase of concentration or uptake? In addition, it is difficult to understand what “Bacitracin and metallic atoms” indicate. Please review the citing articles thoroughly and revise Table 1.

Author Response

Comment:

The authors summarized the recent development of antimicrobial gold nanoclusters and divided them into two categories, namely, small molecules and macromolecules. The citing articles are up to date; however, there are nontrivial number of grammatical errors and typos in this review article. In addition, some information summarized in this review does not clearly convey the ideas from original articles. Please refer to the following suggestions and make proper corrections. The authors should also thoroughly proofread their manuscript before it is ready for publication.

Reply to the reviewer:

We appreciate the comments and suggestions from Reviewer 2. We have corrected the grammatical errors and typos in the revised manuscript. We have also revised the examples used in this review to make their ideas more clearly as raised by Reviewer 2.

Comment:

Ref 8 should come from O'Neill J. Review on Antimicrobial Resistance Antimicrobial Resistance: Tackling a crisis for the health and wealth of nations. London: Review on Antimicrobial Resistance. 2014

Reply to the reviewer:

We have added the reference from O'Neill J in the revised manuscript.

Revision made:

The reference from O'Neill J has been cited as Reference 9.

(O’Neill, J., Review on Antimicrobial Resistance Antimicrobial Resistance: Tackling a crisis for the health and wealth of nations. London: Review on Antimicrobial Resistance. 2014.)

Comment:

Line 42,68,84,141,177,200,228,261,307: included should be including.

Reply to the reviewer:

On Line 42, 68, 84, 141, 177, 200, 228, 261 and 307, the word of “included” has been corrected as “including”.

Comment:

Line 90-92: “The ligands of amino acids, peptides, antibiotics, antibodies, enzymes, DNA and so forth have been demonstrated for the syntheses of AuNCs.” is unclear. It should be rewritten for clarity.

Reply to the reviewer:

The related synthesis approaches for the uses of the ligands of amino acids, peptides, antibiotics, antibodies, enzymes, DNA and so forth to prepare AuNCs have been cited in the References from 57 to 68.

Comment:

Line 100: coast should be cost.

Reply to the reviewer:

Corrected.

Comment:

Line 130: The definition of “S. aureus burden” is unclear.

Reply to the reviewer:

We have rewritten the sentence to make it more clearly. We wrote “The DPAu/AMD improve antibacterial effect by reduction of 65% of S. aureus population compared to that of 42% for the free AMD.”

Comment:

Line 147: should be Au (I).

Reply to the reviewer:

Corrected.

Comment:

Line 149: have “the” highest

Reply to the reviewer:

Corrected.

Comment:

Line 160: “have been show in the Figure 3” should be “have been shown in Figure 3”.

Reply to the reviewer:

Corrected.

Comment:

Line 164: define CFU.

Reply to the reviewer:

The colony-forming unit (CFU) has been added in the revised manuscript.

Comment:

In the caption of Figure 3: Insets of Figure 4A should be Figure 3A.

Reply to the reviewer:

Corrected.

Comment:

Line 192: provide the references for skin infection model.

Reply to the reviewer:

The reference for skin infection model has been cited as Reference 96.

96.  Liu, Y.; Ding, S.; Dietrich, R.; Märtlbauer, E.; Zhu, K., Angew. Chem. Int. Ed. 2017, 56, 1486-1490.

Comment:

Line 204-206: For clarity, it is better to report the charges or Zeta potential for each conjugate as it is important for their antimicrobial activity.

Reply to the reviewer:

The zeta potentials for AuDHMP, AuAMP, AuDAMP and AuAHMP have been added in the revised manuscript. We wrote “The zeta potentials for AuDHMP, AuAMP, AuDAMP and AuAHMP are -38.6±1.8, +33.6±1.4, 37.6±1.1 and 12.7±0.7 mV, respectively.”

Comment:

Line 233: The use of “inference” is unclear.

Reply to the reviewer:

The sentence has been rewritten. We wrote “By the designs of surface ligands, Au25NCs with more negative surface charges on the surface could induce more ROS generation to react with metabolic enzyme of bacteria and then to kill the bacteria (Figure 6) [100,101].”

Comment:

Line 275: temple should be template 

Reply to the reviewer:

Corrected.

Comment:

Line 284 and in Table 1: pentapeptide γ-ECGDADA should be γ-ECGDADA, where DADA means D-alanine-D-alanine.

Reply to the reviewer:

Corrected.

Comment:

Line 285: SGaa is not defined.

Reply to the reviewer:

The sentence has been rewritten. We wrote “Moreover, Li et al. have chosen a pentapeptide γ-ECGDADA (GSHaa) to prepare AuNCs (Au-SGaa, SGaa denotes dehydrogenated GSHaa) and then Au-SGaa have been conjugated with vancomycin (Au-SGaa-Van) [83].”

Comment:

Line 286-289, 320: Bacteria names should be italicized.

Reply to the reviewer:

Corrected.

Comment:

In the caption of Figure 10, the word “synthesis” is misleading. It should be changed from “Illustration of synthesis of” to “Illustration of the preparation of”

Reply to the reviewer:

Corrected.

Comment:

In the conclusion, the authors introduced some current challenges which are not mentioned in the previous sections. It is generally not a good practice to introduce new ideas in the conclusion. I suggest the authors to write more details about the current challenge in a new section. This will add more valuable information to the readers.

Reply to the reviewer:

We have removed the section of “Conclusions” and added the section of “Challenges and Opportunities”.

Challenges and Opportunities

In this mini review, we have summarized recent achievements of AuNCs conjugated with small molecules and macromolecules for the applications as the antimicrobial agents (Table 1). These studies have demonstrated that AuNCs can be the potential antimicrobial agents because of their high biocompatibility, polyvalent effect, easy modification and photothermal stability. Although different AuNCs have been proven as the antimicrobial agents, however, their antimicrobial activities are still needed to be improved. The first challenge to improve the antimicrobial activity of AuNCs is to prepare AuNCs conjugated with antimicrobial surface ligands. The antimicrobial activity can be enhanced by the use of synergistic effect between AuNCs and antimicrobial surface ligands. The second challenge for antimicrobial AuNCs is to increase their cell uptake. With the controls of surface ligands, AuNCs can bear positive charge and negative charge and even to have target-specific property for bacteria to increase the cell uptake of AuNCs. The third challenge for antimicrobial AuNCs is to investigate the details of antimicrobial mechanisms in bacteria. Until now, there are various mechanisms to explain the antimicrobial performance of AuNCs. Therefore, experimental and theoretical investigations of the metabolisms of AuNCs in bacteria are still required for better understanding their antimicrobial activity. Overall, to realize the antimicrobial agents of AuNCs, a lot of works still need to be completed for the improvement of antimicrobial activity of AuNCs to meet the requirement in clinic application. With the extensive investigations, we believe that AuNCs can be applied as the significant antimicrobial agents in clinic in the near future.

Comment:

In Table 1, some of the antimicrobial mechanisms are unclear. For example, does “increase of vancomycin” mean the increase of concentration or uptake? In addition, it is difficult to understand what “Bacitracin and metallic atoms” indicate. Please review the citing articles thoroughly and revise Table 1.

Reply to the reviewer:

We have revised Table 1 to make it more clearly in the revised manuscript.

Round  2

Reviewer 1 Report

The revised version of this mini-review is improved and the frontier at ~900 Da is better justified in terms of the different physico-chemical properties in determining their antimicrobial efficacy.

This paper is thus acceptable for IJMS.

please add Da after molecular weight in Table I to avoid any confusion.